# MASSIVELY MULTILINGUAL SPARSE WORD REPRESENTATIONS

**Gábor Berend** [1,2]
[1] University of Szeged, Institute of Informatics
Szeged, Hungary
[2] MTA-SZTE RGAI, Szeged
Szeged, Hungary
`berendg@inf.u-szeged`

## ABSTRACT

In this paper, we introduce MAMUS for constructing multilingual sparse word representations. Our algorithm operates by determining a shared set of semantic units which get reutilized across languages, providing it a competitive edge both in terms of speed and evaluation performance. We demonstrate that our proposed algorithm behaves competitively to strong baselines through a series of rigorous experiments performed towards downstream applications spanning over dependency parsing, document classification and natural language inference. Additionally, our experiments relying on the QVEC-CCA evaluation score suggests that the proposed sparse word representations convey an increased interpretability as opposed to alternative approaches. Finally, we are releasing our multilingual sparse word representations for the 27 typologically diverse set of languages that we conducted our various experiments on.

## 1 INTRODUCTION

Cross-lingual transferability of natural language processing models is of paramount importance for low-resource languages which often lack a sufficient amount of training data for various NLP tasks. A series of attempts have been made to remedy the shortage of labeled training data. Both part-of-speech tagging (Fang & Cohn, 2017; Zhang et al., 2016; Haghighi & Klein, 2006; Gouws & Søgaard, 2015; Kim et al., 2015; Agić et al., 2015) and dependency parsing (Guo et al., 2015; Agić et al., 2016) have been investigated from that perspective. The mapping of distributed word representations of low-resource languages to the embedding space of a resource-rich language makes NLP models trained on the source language directly applicable for texts in resource-scarce languages.

Overcomplete word representations aim at expressing low dimensional distributed word representations as a sparse linear combination of an overcomplete set of basis vectors. Using sparse word representations has not only been reported to give better performance compared to dense representations for certain problems (Yogatama et al., 2015; Faruqui et al., 2015; Sun et al., 2016; Berend, 2017), but it is also argued to provide increased interpretability (Murphy et al., 2012; Vyas & Carpuat, 2016; Park et al., 2017; Subramanian et al., 2018). Such bilingual sparse representations have straightforward benefits as any machine learning model that is trained on the labeled training data using the sparse representation of some resource-rich source language can be used directly and interchangeably for texts written in some target language. This way we can enjoy the benefits of sparsity such as smaller and more interpretable models.

In this work, we propose a new algorithm for determining multilingual sparse word representations in which words with similar meaning across different languages are described with similar sparse vectors. Our work has multiple advantageous properties compared to previous similar attempts (Vyas & Carpuat, 2016; Upadhyay et al., 2018). Firstly, our algorithm naturally fits multilingual setting, i.e. it is not limited to pairs of languages, but is capable of determining cross-lingual sparse embeddings for multiple languages.

Our algorithm differs from previous approaches in that we obtain target language sparse word representations via a series of convex optimization problems with a substantially reduced parameter set,

whereas previous solutions perform non-convex optimization for orders of more parameters. The model formulation we propose not only speeds up computation drastically, but our empirical results also suggest that the proposed sparse word vectors perform better when employed in various extrinsic tasks, i.e. dependency parsing, document classification and natural language inference. We make our sparse embeddings for 27 languages and the source code that we used to obtain them publicly available at `https://github.com/begab/mamus`.

## 2 MAMUS – THE PROPOSED ALGORITHM

In order to facilitate the introduction of the MAssively MUltilingual Sparse (dubbed as MAMUS) word representations, we first introduce our notation. In what follows, we denote dense word embedding matrices of the source and target languages as $S \in \mathbb{R}^{m \times |V_s|}$ and $T \in \mathbb{R}^{n \times |V_t|}$, respectively, with $V_s$ and $V_t$ indicating the vocabulary of the source and target languages. The $m = n$ condition does not have to hold in general, however, it is going to be the case throughout the paper. We shall denote some symbolic word form as $x$ and its corresponding distributed vectorial representation in boldface, i.e. $\mathbf{x}$.

### 2.1 PREPROCESSING OF INPUT WORD EMBEDDINGS

Our algorithm takes as input a pair of "traditionally" trained distributed dense word embeddings such as Glove (Pennington et al., 2014), word2vec (Mikolov et al., 2013a) or fasttext (Bojanowski et al., 2017). It is important to note that we do not assume the existence of parallel text, meaning that we can train the input word embeddings in total isolation from each other.

We transform the dense input embedding matrices so that the word vectors comprising it have unit norm, as it is frequently met in the literature (Xing et al., 2015; Artetxe et al., 2016; 2017). This preprocessing step ensures that the dot product of word embeddings equals their cosine similarity. Unit normalization of word embeddings also makes cross-lingual comparison more natural, since all embeddings have identical length, irrespective of the language they belong to.

### 2.2 MAPPING OF WORD EMBEDDINGS

In order to align independently trained word representations, we learn a linear mapping $W$. The mapping is expected to bring target language word vectors close to their semantically equivalent counterparts in the source embedding space.

As proposed by Mikolov et al. (2013b) such $W$ can be simply defined by minimizing $\sum_{i=1}^{l} \|\mathbf{s_i} - W\mathbf{t_i}\|$, with $\{(s_i, t_i)\}_{i=1}^{l}$ being a set of word pairs which are cross-lingual equivalents of each other. If we construct matrices $S_o = [\mathbf{s_i}]_{i=1}^{l}$ and $T_o = [\mathbf{t_i}]_{i=1}^{l}$ by stacking the embeddings of translation pairs, we can express the solution as $S_o T_o^+$, with $T_o^+$ denoting the Moore-Penrose pseudoinverse of $T_o$.

Multiple studies have argued that ensuring $W$ to be orthonormal can significantly improve the quality of the mapping of word embeddings across languages (Smith et al., 2017; Xing et al., 2015; Artetxe et al., 2016; Hamilton et al., 2016). Finding the optimal orthonormal $W$ can be viewed as an instance of the orthogonal Procrustes problem (Schönemann, 1966) which can be solved by $W_\perp = UV$, with $U$ and $V$ coming from the singular value decomposition of the matrix product $S_o^\intercal T_o$. By applying an isometric transformation, we ensure that transformed embeddings preserve their norm.

### 2.3 SPARSE CODING OF THE EMBEDDING SPACES

Sparse coding decomposes a $X \in \mathbb{R}^{m \times v}$ matrix of signals (word embeddings in our case) into the product of a dictionary matrix $D \in \mathbb{R}^{m \times k}$ and a matrix of sparse coefficients $\alpha \in \mathbb{R}^{k \times v}$, where $k$ denotes the number of basis vectors to be employed. The columns of $D$ form an overcomplete set of basis vectors and the sparse nonzero coefficients in the $i^{th}$ column of $\alpha$ indicate which column

vectors from $D$ should be incorporated in the reconstruction of $\mathbf{x}_i$. We minimize the objective

$$\min_{D \in \mathcal{C}, \alpha \in \mathbb{R}_{\geq 0}^{k \times |v|}} \frac{1}{2} \|X - D\alpha\|_F^2 + \lambda\|\alpha\|_1, \tag{1}$$

where $\mathcal{C}$ denotes the convex set of matrices with column norm at most 1 and the sparse coefficients in $\alpha$ are required to be non-negative. Ensuring that all the coefficients in $\alpha$ are non-negative makes their cross-lingual comparison more natural, as the signs of sparse word coefficients cannot mismatch. Additionally, non-negativity has been reported to provide increased interpretability (Murphy et al., 2012). We used the SPAMS library (Mairal et al., 2009) for calculating $D$ and $\alpha$.

We can perform sparse coding as defined in (1) for the independently created source and target embedding matrices $S$ and $T$, obtaining decompositions $S \approx D_s \alpha_s$ and $T \approx D_t \alpha_t$. If we do so, however, sparse word representations extractable from matrices $\alpha_s$ and $\alpha_t$ are not comparable to any extent due to the fact that $S$ and $T$ are decomposed independently of each other.

We propose to first solve a single instance of the non-convex dictionary learning problem as defined in (1) for the source language alone as if our goal was to obtain monolingual sparse word representations. After determining $D_s$, we can apply this dictionary matrix to find sparse coefficient matrices for the isometrically transformed embeddings of all the target languages, hence solving a much simpler complex optimization problem with reasonably fewer parameters in the case of target languages of the form

$$\min_{\alpha \in \mathbb{R}_{\geq 0}^{k \times |v|}} \frac{1}{2} \|W_\perp X - D_s \alpha\|_F^2 + \lambda\|\alpha\|_1, \tag{2}$$

with $W_\perp$ denoting the isometric transformation specific to a given source-target language pair. We summarize the working mechanisms of MAMUS in Algorithm 1.

---

**Algorithm 1** Pseudocode of MAMUS

---

**Require:** source and target embeddings $S \in \mathbb{R}^{m \times |V_s|}, T_1 \in \mathbb{R}^{m \times |V_{t_1}|}, \ldots, T_N \in \mathbb{R}^{m \times |V_{t_N}|}$
  semantically equivalent word pairs $\{(s_1^{(i)}, t_1^{(i)})\}_{i=1}^{l_1}, \ldots, \{(s_N^{(i)}, t_N^{(i)})\}_{i=1}^{l_N}$
**Ensure:** Sparse representation matrices $(\alpha_s, \alpha_{t_1}, \ldots, \alpha_{t_N})$
  **procedure** MAMUS$(S, T)$
    $S \leftarrow \text{UNITNORMALIZE}(S)$
    $D_s^*, \alpha_s^* \leftarrow \underset{\alpha, D}{\arg\min} \|S - D\alpha\|_F + \lambda\|\alpha\|_1$
    **for** $k \leftarrow 1$ to $N$ **do**
      $T_k \leftarrow \text{UNITNORMALIZE}(T_k)$
      $W_\perp^* \leftarrow \underset{W^\intercal W = I}{\arg\min} \sum_{i=1}^{l_k} \|\mathbf{s_k^{(i)}} - W\mathbf{t_k^{(i)}}\|$
      $\alpha_{t_k} \leftarrow \underset{\alpha \in \mathbb{R}_{\geq 0}}{\arg\min} \|W_\perp^* T_k - D_s^* \alpha\|_F + \lambda\|\alpha\|_1$
    **end for**
    **return** $\alpha_s, \alpha_{t_1}, \ldots, \alpha_{t_N}$
  **end procedure**

---

### 2.4 Decisive differences to BISPARSE

The goal of the BISPARSE (Vyas & Carpuat, 2016) is to determine such sparse word representations that behave similarly for a pair of languages. We subsequently juxtapose the conceptual differences between BISPARSE and MAMUS both from theoretical and practical perspectives.

BISPARSE (Vyas & Carpuat, 2016) determines sparse bilingual word representations by solving

$$\min_{\substack{D_s, D_t \in \mathbb{R}^{m \times k}, \\ \alpha_s \in \mathbb{R}_{\geq 0}^{k \times |V_s|}, \\ \alpha_t \in \mathbb{R}_{\geq 0}^{k \times |V_t|}}} \frac{1}{2}\Big(\|X_s - D_s\alpha_s\|_F^2 + \|X_t - D_t\alpha_t\|_F^2 + \lambda_x \sum_{i=1}^{|V_s|} \sum_{j=1}^{|V_t|} M_{ij}\|\boldsymbol{\alpha_s^{(i)}} - \boldsymbol{\alpha_t^{(j)}}\|_2^2\Big) + \lambda_t\|\alpha_t\|_1 + \lambda_s\|\alpha_s\|_1,$$

$$\tag{3}$$

where $M$ is a pairwise similarity matrix between the source and target language word pairs, $\lambda_s, \lambda_t$ and $\lambda_x$ denotes the regularization coefficients for the source, target language and the cross-lingual loss, respectively. Finally, $\boldsymbol{\alpha_s^{(i)}}$ and $\boldsymbol{\alpha_t^{(j)}}$ refers to the $i^{th}$ and $j^{th}$ columns of matrices $\alpha_s$ and $\alpha_t$, respectively.

As a first notable difference, BISPARSE operates by employing an explicit $|V_s| \times |V_t|$ pairwise similarity matrix, whereas MAMUS relies on a list of word translation pairs, which is used to rotate all the target language embeddings into the embedding space of the source language. BISPARSE hence requires an additional hyperparameter which controls for the strength of the cross-lingual regularization (cf. $\lambda_x$). According to our experiments choosing this hyperparameter is a crucial factor in employing BISPARSE.

The objective in (3) reveals that the original BISPARSE model optimizes the source and target word representations jointly, i.e. a separate source language representation is created whenever applied to a new target. This behavior of BISPARSE limits its application for a single pair of languages, instead of the more general multilingual setting, when our goal is to represent more than two languages at a time. As the default behavior of BISPARSE hampers truly multilingual application, we adopt it similar to MAMUS, i.e. we first train sparse word representations for the source language by solving

$$\min_{D_s \in \mathbb{R}^{m \times k}, \alpha_s \in \mathbb{R}_{\geq 0}^{k \times |V_s|}} \frac{1}{2} \|X_s - D_s \alpha_s\|_F^2 + \lambda_s \|\alpha_s\|_1, \tag{4}$$

with the source language being chosen as English. Once the dictionary matrix of semantic atoms $(D_s^*)$ and the sparse coefficients $(\alpha_s^*)$ are calculated for the source language, we next solve

$$\min_{D_t \in \mathbb{R}^{m \times k}, \alpha_t \in \mathbb{R}_{\geq 0}^{k \times |V_t|}} \frac{1}{2} \|X_t - D_t \alpha_t\|_F^2 + \lambda_t \|\alpha_t\|_1 + \frac{1}{2} \lambda_x \sum_{i=1}^{|V_s|} \sum_{j=1}^{|V_t|} M_{ij} \|\boldsymbol{\alpha_s^{*(i)}} - \boldsymbol{\alpha_t^{(j)}}\|_2^2. \tag{5}$$

Identical to BISPARSE (Vyas & Carpuat, 2016), we solved the modified optimization problems in (4) and (5) relying on the efficient Forward-Backward Splitting Solver implementation of the FASTA framework (Goldstein et al., 2014; 2015).

To summarize, the way BISPARSE works in its original formulation is that it creates sparse word representations for pairs of languages, both for the source and target language. One compelling characteristic of MAMUS, however, is that it learns the $D_s$ and $\alpha_s$ parameters independent from the target embedding space, thus the same decomposition of source language embeddings can be utilized for multiple target languages.

This makes our algorithm conveniently applicable to an arbitrary number of languages. Furthermore, by using the dictionary matrix $D_s^*$ for all the target languages, MAMUS not only involves optimization over a substantially reduced set of parameters, it additionally enjoys the ease of solving a series a convex optimization problems. The model formulation of BISPARSE on the other hand is such, that it requires to solve a series of highly non-convex optimization problems for each of the target languages, since it requires the optimization over $D_t$ as well. This property is also true for the modified version of BISPARSE which is adapted for multilingual usage. We will refer to this adapted version of BISPARSE as MULTISPARSE for the rest of the paper.

## 3 EXPERIMENTS

Our primary source for evaluating our proposed representations is the massively multilingual evaluation framework from (Ammar et al., 2016b), which also includes recommended corpora to be used for training word representations for more than 70 languages. All the embeddings used in our experiments were trained over these recommended resources, which is a combination of the Leipzig Corpora Collection (Goldhahn et al., 2012) and Europarl (Koehn, 2005).

For 11 languages (bg, cs, da, de, el, es, fi, fr, hu, it, sv) – with ample parallel text to English – bilingual dictionaries are also released as part of the evaluation framework. For the remaining languages, Ammar et al. (2016b) released dictionaries that were obtained by translating the 20k most common English words with Google Translate. All the word representations involved in our

Table 1: QVEC-CCA results on the English dev set from (Ammar et al., 2016b) as a function of $\lambda$.

| $\lambda$ | 0.1 | 0.2 | 0.3 | 0.4 | 0.5 |
|---|---|---|---|---|---|
| English QVEC-CCA (development set) | **0.811** | 0.806 | 0.804 | 0.803 | 0.799 |
| Avg. nonzero coefficients per word form | 22.9 | 9.1 | 5.2 | 3.4 | 2.3 |

experiments were trained over the previously introduced resources from (Ammar et al., 2016b) in order to ensure the comparability of the different approaches.

We trained fasttext-CBOW (Bojanowski et al., 2017) dense embeddings as inputs to our experiments. We simply used the default settings of fasttext for training, meaning that the original dense word representations were 100 dimensional. We conducted additional experiments with different input embeddings (word2vec and Glove) and also with different dimensionality (300), however, we omit these results for brevity as the general trends we observed were very similar. We set the number of semantic atoms in the dictionary matrix $D$ consistently as $k = 1200$ throughout all our experiments.

## 3.1 MONOLINGUAL EXPERIMENTS

We first performed monolingual experiments in order to investigate the effects of choosing different regularization coefficients $\lambda$, controlling the sparsity and the quality of sparse word representations.

Table 1 reports the quality of our monolingual representations evaluated by the QVEC-CCA (Tsvetkov et al., 2016) correlation-based intrinsic evaluation metric, which was proposed as an improvement over the QVEC evaluation technique (Tsvetkov et al., 2015). The goal for both QVEC and QVEC-CCA is to quantify the extent to which embedding dimensions can be aligned to human-interpretable concepts, such as word supersenses. The evaluation environment we utilize includes supersense tag annotations for Danish (Alonso et al., 2015; Martinez Alonso et al., 2016), English (Miller et al., 1993) and Italian (Montemagni et al., 2003).

During the monolingual experiments, we were solely focusing on the development set for English to set the hyperparameter controlling the sparsity of the representations. Table 1 additionally contains the average number of nonzero coefficients yielded by the different regularization coefficients.

Based on our monolingual evaluation results from Table 1, we decided to fix the regularization coefficient for MAMUS at $\lambda = 0.1$ for all of our upcoming mutlilingual experiments. As our choice for the value of $\lambda$ was based on the performance achieved by MAMUS on the development set of a single language and a single evaluation criterion, we chose this hyperparameter without the risk of overfitting to the test data regarding any of the multilingual evaluation scenarios.

In the case of MULTISPARSE, we managed to obtain a similar performance and average sparseness (55.2 nonzero coefficient per word form) when utilizing $\lambda_s = 2$. During our multilingual experiments, we set the target language regularization coefficient identically, i.e. $\lambda_t = 2$. For the cross-lingual regularization term of MULTISPARSE, we chose $\lambda_x = 5$ based on our preliminary investigation based on the remaining two languages we had supersense tag annotations for, i.e. Danish and Italian. The above set of hyperparameters resulted in 41.0 and 44.3 nonzero coefficients per word form on average, respectively.

### 3.1.1 MONOLINGUAL EVALUATION OF BISPARSE AND MULTISPARSE REPRESENTATIONS

We conducted an additional monolingual QVEC-CCA evaluation on Danish, English and Italian. In this experiment we compared the results of sparse word representations obtained by the vanilla BISPARSE formulation and its multilingual extension MULTISPARSE. This way we can assess the effects of multilingual training as opposed to bilingual one.

The QVEC-CCA scores we obtained for the standard BISPARSE representations were 0.602, 0.789/0.786 and 0.585 for Danish, English and Italian, respectively. We report two English results (0.789/0.786) as BISPARSE provided us with two distinct representations for English depending on the target language it was jointly trained with. MULTISPARSE representations on the other hand obtained QVEC-CCA scores of 0.612, 0.808 and 0.596 for Danish, English and Italian, respectively.

Our monolingual QVEC-CCA results suggest that the multilingual training objective performs comparably (or even better) to the bilingual one employed in BISPARSE. A possible explanation for this is that BISPARSE tries to optimize a more difficult objective – as it involves the joint optimization of two non-convex problems with an increased number of parameters – hence finding a good solution is more difficult for BISPARSE.

## 3.2 MULTILINGUAL EXPERIMENTS

In the recent survey, Ruder et al. (2019) recommend to evaluate cross-lingual word embeddings on an intrinsic task and at least one downstream task besides document classification. Following this suggestion, we measure the quality of word representations based on QVEC-CCA – similar to our monolingual evaluation – and also report their performance scores in cross-lingual document classification (CLDC), cross-lingual dependency parsing and cross-lingual natural language inference (XNLI). Results from this point on are all obtained with the exact same hyperparameters as introduced before and on the test sets of the various tasks, unless stated otherwise.

### 3.2.1 BASELINE DENSE CROSS-LINGUAL REPRESENTATIONS

Before delving into the multilingual experimental results, we introduce those additional dense distributed cross-lingual representations that we used in our evaluation. These representations were trained over the same corpora and translation lists introduced earlier.

multiCluster (Ammar et al., 2016b) builds a graph from word forms across multiple languages with edges going between translated word pairs. Clusters are then formed based on the connected components of the graph. Word forms are finally replaced by the random identifier of the cluster they belong to and a skip-gram model is trained for the corpus obtained this way.

multiCCA (Ammar et al., 2016b) is an extension of the approach introduced by Faruqui & Dyer (2014). It seeks a linear operator which projects pre–trained word embeddings of some language to the embedding space of English such that the correlation between the representation of translated word pairs is maximized.

Even though we trained our representations and further baselines on the same corpora and word translation lists as the multiCluster and multiCCA were built on, minor differences in the learned vocabularies exists (even between multiCluster and multiCCA) which are likely due to small differences in preprocessing. For this reason, pre–trained multiCluster and multiCCA allow for a less stringent comparison with the remaining representations, which were trained by ensuring to have identical vocabularies to the one employed in our approach. With this caution in mind, we still think it is useful to report evaluation results of these pre-trained embeddings as well.

MUSE (Lample et al., 2018) is capable of training dense cross-lingual word embeddings in both unsupervised and supervised manner. We experiment with the supervised variant of MUSE which requires translated word pairs as input similar to MAMUS. MUSE also incorporates a method for iteratively refining the list of word translation pairs that we relied on using its default parametrization.

### 3.2.2 MULTILINGUAL EXPERIMENTAL RESULTS

**QVEC-CCA** We next assess the quality of the different multilingual word representations according to the QVEC-CCA evaluation score. As mentioned earlier, the evaluation framework introduced by Ammar et al. (2016b) provides supersense tagging for Danish, English and Italian. In Table 2a, we report the individual evaluation scores over the three subspaces and their combination, from which we can conclude that MAMUS has a clear advantage over all the alternative word representation regarding this interpretability–oriented evaluation metric.

**Downstream evaluations from Ammar et al. (2016b)** We also performed a 4-class cross-lingual document classification (CLDC) on newswire texts originating from the RCV corpus (Lewis et al., 2004) over 7 languages (da, de, en, es, fr, it, sv). The model was simultaneously tuned on the training sections of the different languages as implemented in (Ammar et al., 2016b). Table 2b lists classification accuracies of the different methods.

Table 2: Intrinsic (a) and extrinsic (b) evaluations from the test bed from (Ammar et al., 2016b).

(a) Intrinsic QVEC-CCA evaluation scores of various cross-lingual word representations.

(b) Downstream evaluation results as average accuracy for CLDC and UAS for dependency parsing.

| | da | en | it | {da,en,it} | | CLDC | dependency parsing |
|---|---|---|---|---|---|---|---|
| multiCluster | 0.475 | 0.539 | 0.468 | 0.372 | | 90.79 | 61.39 |
| multiCCA | 0.501 | 0.635 | 0.510 | 0.415 | | **92.18** | 62.82 |
| MUSE | 0.343 | 0.393 | 0.338 | 0.294 | | 87.34 | **64.47** |
| MULTISPARSE | 0.612 | 0.808 | 0.596 | 0.480 | | 86.45 | 59.20 |
| MAMUS | **0.632** | **0.819** | **0.620** | **0.503** | | 91.84 | 63.53 |

Regarding our evaluation towards dependency parsing, we evaluated the transition-based stack-LSTM parser from (Ammar et al., 2016a) over the Universal Dependencies v1.1 treebanks (Agić et al., 2015) covering 18 languages (bg, cs, da, de, el, en, es, eu, fa, fi, fr, ga, he, hr, hu, id, it, sv). Even though the parser is capable of incorporating additional features besides word embeddings, this capability of the parser was disabled, so that the effects of employing different word representations can be assessed on their own. The performance metric for parsing is reported in Table 2b as unlabeled attachment score (UAS).

Table 2b illustrates that the performance obtained by dense representations varies largely between downstream tasks. The performance of MUSE, for instance, is the best when evaluated on dependency parsing, however, it has the second lowest overall accuracy for CLDC. Evaluation scores for those models that are based on MAMUS representations rank second on both downstream evaluation tasks with a minor performance gap to the best results obtained by different dense representations on the two tasks.

**Natural Language Inference** In order to assess the capabilities of the different representations towards natural language inference (NLI) in a multilingual setting, we also performed evaluation towards the XNLI dataset (Conneau et al., 2018). XNLI covers 15 languages and it can be viewed as a multilingual extension of the multiNLI (Williams et al., 2018) dataset. The task in XNLI and multiNLI is to categorize sentence pairs – comprising of a premise ($p$) and a hypothesis ($h$) sentence – whether the relation between $p$ and $h$ is entailing, contradictory or neutral.

We implemented a simple multilayer perceptron in PyTorch v1.1 (Paszke et al., 2017) with two hidden layers employing ReLU nonlinearity. The MLP uses the categorical cross-entropy for loss function, which was optimized by Adam (Kingma & Ba, 2014). Based on the differently constructed sparse and dense word representations, we train five different NLI models based on the English multiNLI dataset and report average classification accuracies as the performance score. The five models only differed in the random initialization of the parameters to account for the potential variability in model performances.

For a pair of premise and hypothesis sentence representation pair $(\boldsymbol{p}, \boldsymbol{h})$, the input to the MLP gets determined according to
$$[\boldsymbol{p}; \boldsymbol{h}; \boldsymbol{p} - \boldsymbol{h}; \boldsymbol{p} \odot \boldsymbol{h}],$$
similar to (Williams et al., 2018) with ; standing for vector concatenation and $\odot$ denoting element-wise multiplication. The vectorial representations for the individual sentences were also obtained in an identical way to the approach (dubbed as X-CBOW) in (Conneau et al., 2018), i.e. we took the mean of the vectorial representations of the words comprising a sentence.

We report the average accuracy achieved by the different word representations for multiNLI (Table 3a) and the 15 languages of XNLI (Table 3b). Although the training corpora compiled by Ammar et al. (2016b) contains datasets for all the XNLI languages, the authors did not release any pre-trained multiCluster and multiCCA embeddings for Hindi (hi), Thai (th), Urdu (ur) and Vietnamese (vi). As a result, we do not report multiCluster and multiCCA results for these four languages.

We can see in Table 3a that – according to the evaluation conducted on the development set of multiNLI – multiCCA performed the best (similar to CLDC), and the English subspace of the MAMUS representations achieved the second best results. Table 3b represents the average performance

Table 3: Averaged multiNLI and XNLI performances of 5 independent MLPs.

(a) Average multiNLI matched dev set accuracy of the five models used during XNLI evaluation.

| multiCluster | multiCCA | MUSE | MULTISPARSE | MAMUS |
|---|---|---|---|---|
| 57.74 | 63.43 | 57.10 | 60.22 | 61.43 |

(b) XNLI results obtained. Results in bold are the best for a given language, underlined scores indicate the second best resutls.

| | ar | bg | de | el | en | es | fr | ru | sw | tr | zh | hi | th | ur | vi |
|---|---|---|---|---|---|---|---|---|---|---|---|---|---|---|---|
| multiCluster | 33.36 | **49.53** | **50.00** | 49.52 | 58.84 | 48.86 | 48.46 | 39.47 | 35.91 | 36.97 | 36.05 | — | — | — | — |
| multiCCA | 38.85 | 40.59 | 34.50 | 42.70 | **65.43** | 36.16 | 33.83 | 41.25 | **38.31** | 38.82 | **45.36** | — | — | — | — |
| MUSE | **42.83** | 45.64 | 41.60 | 46.24 | 58.39 | 44.53 | 43.64 | **44.70** | 37.33 | 42.21 | 44.57 | 38.98 | 33.50 | 38.57 | **35.92** |
| MULTISPARSE | 33.10 | 35.04 | 38.17 | 33.33 | 60.10 | 40.48 | 40.92 | 33.34 | 36.20 | 37.78 | 33.23 | 35.40 | 33.06 | 34.18 | 34.63 |
| MAMUS | 42.67 | 47.88 | 44.27 | **49.58** | **61.53** | **51.19** | **48.93** | 43.89 | 37.09 | **47.24** | 44.73 | **41.78** | **34.59** | **42.00** | 34.94 |

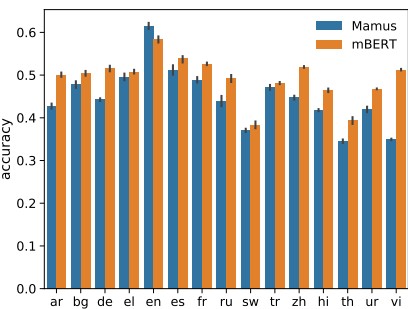

Figure 1: Comparison of the accuracies over the 15 languages of the XNLI datasets.

of five models that are purely trained on the English multiNLI training data relying on the various multilingual word representations upon evaluation. It can be seen in Table 3b that – other than for Swahili – the models trained on the English MAMUS representations perform either the best or the second best 14 out of the 15 languages.

In order to account for the recent dominance of contextualized word representations, we additionally compared the application of MAMUS to that of multilingual BERT (Devlin et al., 2019) when utilized for XNLI. We employed multilingual BERT (mBERT) representations from Wolf et al. (2019) identically to how we used the non-contextualized word representations previously. That is, we took the average of the contextualized representations that we derived from the last layer of mBERT and trained 5 MLPs with different weight initializations. Similar to our previous experiments, we conducted evaluation in the zero-shot setting for the 15 XNLI languages, i.e. by solely using multiNLI for training our models. The comparative results of the trained MLPs can be seen in Figure 1.

Figure 1 reveals that the MLPs trained on top of MAMUS representations are capable of achieving more than 90% relative performance on average to those MLPs which rely on mBERT. Indeed, the average performance of the MLPs over all the languages and random initializations is 49.3 for mBERT, while MAMUS-based MLPs had an average accuracy of 44.8. We thus think that MAMUS can be a viable alternative to mBERT in multilingual settings when a small fraction of accuracy can be sacrificed for obtaining a model in which the calculation of sentence representations is substantially cheaper (cf. the computational requirements of averaging static word embeddings and contextual embeddings which require a forward pass in a transformer architecture).

Interestingly, when inspecting results for English in Figure 1, we can see that MLPs taking MAMUS representations as input actually perform better than those utilizing mBERT by a fair margin (61.5 versus 58.3). This could imply that our algorithm has the potential to transform static dense embeddings in a way which makes them as useful as contextualized mBERT representations for certain applications. Note that this hypothesis would require further investigation, that is left for future work.

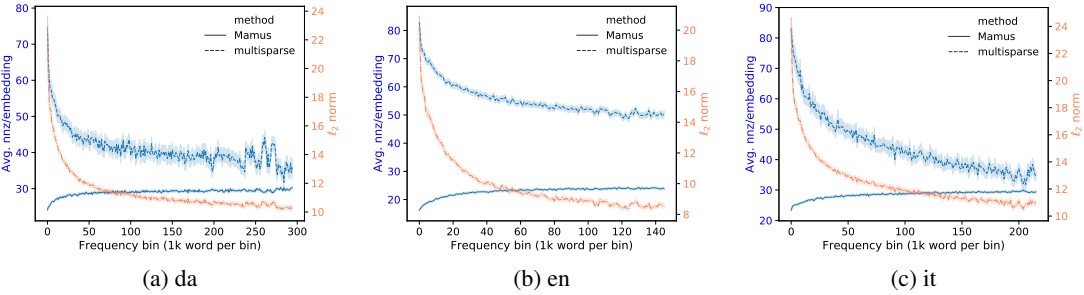

(a) da          (b) en          (c) it

Figure 2: The sparsity structure of the learned representations for the languages covered in the multilingual QVEC-CCA evaluation as a function of the frequency of word forms. Plots also include the frequency-binned average norms of the dense input embeddings. The orange curve indicates the norms of the embeddings per frequency bins, the blue are for the number of nonzero coefficients per word forms.

### 3.2.3 ANALYSIS OF THE SPARSITY STRUCTURE

As a final assessment of the sparse word representations, we characterize their number of nonzero coefficients as a function of the frequency of the words they correspond to.

From a human cognition point of view, we can argue that frequency can be a good proxy to the specialization in the meaning of a word (Caraballo & Charniak, 1999). Words with high frequency, e.g. *car*, *dog* and *newspaper* tend to refer to easily definable concepts, whereas less frequent words, such as *gymnosperms* or *samizdat* have a more complex – hence more difficult to describe – meaning.

In terms of sparse coding, this could be reflected by the fact that words with more complex meaning would rely on more semantic atoms from the dictionary matrix $D$. Encoding less frequent words with more bits can also be motivated from an information theoretic point of view.

In Figure 2, we plot the average number of nonzero elements over the entire vocabulary of the Danish, English and Italian subspaces (grouped into bins of 1,000 words). We omit similar figures for the rest of the languages for space considerations.

Figure 2 reveals that the sparsity structures induced by the different approaches differ substantially. MAMUS tends to behave more stable across the languages, i.e. it tends to assign a similar amount of nonzero coefficients for words in different languages that belong to the same frequency bin. The number of nonzero coefficients per a word form determined by MULTISPARSE, however behaves less predictably.

Figure 2 also illustrates that the sparsity structure of MULTISPARSE is highly influenced by the norms of the dense input embeddings, which are known to be typically higher for more common word forms (Turian et al., 2010). Representations determined by MAMUS, however, tend to assign more nonzero coefficients to less frequent – hence arguably more specialized – word forms.

## 4 RELATED WORK

A common technique employed to overcome the absence of labeled training data is to apply cross-lingual projections to port the missing linguistic annotation for low-resource target languages (Yarowsky & Ngai, 2001; Das & Petrov, 2011; Täckström et al., 2013; Agić et al., 2015; Agić et al., 2016). Such projections are often determined by word alignment algorithms which imply that these kind of approaches inherently require substantial amounts of parallel text.

There has been a series of research conducted for handling low-resource languages. A dominant approach is to rely on some form of dictionary between a low-resource and a resource-rich language which can be used to perform canonical correlation analysis (Kim et al., 2015), directly incorporated into the learning procedure of word embeddings (Gouws & Søgaard, 2015) or use it in a post-hoc manner to map independently trained word embeddings (Fang & Cohn, 2017; Zhang et al., 2016).

Model transfer techniques, in which an initial model is trained on a resource-rich language and adapted for a target language in a semi-supervised or unsupervised manner, are also popular (Fang & Cohn, 2017; Zhang et al., 2016).

There is massive research interest in transforming word representations such that they become comparable across languages with little or no supervision (Zhang et al., 2017b;a; Artetxe et al., 2017; Smith et al., 2017; Lample et al., 2018; Joulin et al., 2018). There has been a myriad of further techniques introduced for determining cross-lingual distributed word representations (Klementiev et al., 2012; Hermann & Blunsom, 2014; Faruqui & Dyer, 2014; Huang et al., 2015; Luong et al., 2015; Gouws et al., 2015; Vulić & Moens, 2015; Ammar et al., 2016b), *inter alia*. The proposed approaches differ widely in the assumptions they make regarding the amount of available parallel or comparable data for determining bilingual word embeddings. Upadhyay et al. (2016) and Ruder et al. (2019) provide extensive overviews on the available approaches.

Contrary to these methods, our algorithm relates to the line of research focusing on sparse word representations. Such sparse word representations have been shown to outperform dense word embeddings in monolingual settings (Murphy et al., 2012; Yogatama et al., 2015; Faruqui et al., 2015; Berend, 2017; Sun et al., 2016; Subramanian et al., 2018).

Vyas & Carpuat (2016) proposed BISPARSE for obtaining comparable sparse word representations in a bilingual setting. As mentioned earlier, BISPARSE needs to be modified in order to be applicable for obtaining multilingual representations. Perhaps even more importantly, even when applied in the bilingual case, BISPARSE needs to perform a highly non-convex optimization problem and for a substantially larger number of parameters in contrast to MAMUS. Upadhyay et al. (2018) extends (Vyas & Carpuat, 2016) by additionally incorporating dependency relations into sparse coding.

## 5 CONCLUSIONS

In this paper we introduced MAMUS for determining cross-lingually comparable sparse word representations. Our model formulation allowed us to solve a series of convex optimization problems per each target language, which resulted in a more favorable overall training time (4 hours for MAMUS as opposed to 300 hours when using MULTISPARSE) over the 27 languages we conducted our evaluations on. Finally, we make our multilingual sparse embeddings for 27 languages publicly available at `https://github.com/begab/mamus`.

## ACKNOWLEDGEMENTS

This research was partly funded by the project "Integrated program for training new generation of scientists in the fields of computer science", no EFOP-3.6.3-VEKOP-16-2017-0002, supported by the EU and co-funded by the European Social Fund. This work was in part supported by the National Research, Development and Innovation Office of Hungary through the Artificial Intelligence National Excellence Program (grant no.: 2018-1.2.1-NKP-2018-00008). This research was supported by grant TUDFO/47138-1/2019-ITM of the Ministry for Innovation and Technology, Hungary.

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
