# OpenReview forum: "Massively Multilingual Sparse Word Representations"
_ICLR.cc/2020/Conference — Accept (Poster)_

### Official Review · AnonReviewer2 · 2019-10-23
**Official Blind Review #2**

**Rating:** 8

**Review:**

This paper proposes a method to generate sparse multilingual embeddings. The key idea is to build only one set of source basis embeddings, and then represent all multilingual embeddings as a linear combination of these source embeddings. I felt the paper is a nice extension of the Vyas 2016 paper. Compared to existing approaches, their method will be faster to train and will need less data (particularly useful for low resource languages). Since I am less aware of work in this area, I cannot comment on whether the evaluation is complete. Particularly, I wonder if there is a qualitative way to show interpretability of the sparse vectors created by the method. We currently only have QVEC-CCA numbers to judge interpretability.  A few more suggestions:

1. I got confused by the sentence ".. over a reduced number of parameters for each target language as it treats D_s as D_t .". Things got clear from the equations, but will be good to fix.

2. Figure 1 was very difficult to understand. Ideally your caption should be enough to understand the figure.

**Experience Assessment:**

I have read many papers in this area.

**Review Assessment: Checking Correctness Of Derivations And Theory:**

I assessed the sensibility of the derivations and theory.

**Review Assessment: Checking Correctness Of Experiments:**

I carefully checked the experiments.

**Review Assessment: Thoroughness In Paper Reading:**

I read the paper thoroughly.

---

> ### Author Response · Authors · 2019-11-12
> **Answers to Reviewer #2**
>
> We would like to thank the reviewer for the feedbacks provided.
>
> We agree with the review that the investigation of the actual interpretability of the word representations besides their QVEC-CCA scores could provide additional useful insights.
> Conducting such experiments (e.g. performing Word intrusion detection) was currently beyond the scope of the paper, however, marks an interesting future path.
>
> Based on the suggestions on how to make the paper more easily accessible, we submitted an updated version of the paper.

---

### Official Review · AnonReviewer3 · 2019-10-24
**Official Blind Review #3**

**Rating:** 8

**Review:**

The paper proposes a new approach for generating multilingual sparse representations.
For generating such representations, the proposed approach solves a series of convex optimization problems, serially for each language. Compared to previous work for generating sparse cross-lingual representations which is applicable to a pair of languages, the proposed approach is applicable to an arbitrary number of languages.

The paper argues that these sparse representations can lead to better performance for downstream tasks and interpretability. This is demonstrated using experiments on QVEC-CCA (for interpretability analysis), NLI, cross-lingual document classification, and dependency parsing (downstream tasks).

Overall, the approach is well-motivated and performs well empirically. The experimental setup is also described in detail.

Minor - I did not understand the benefit of MAMUS being "stable" across languages, as argued from Fig 1? The use of the word "cognitively" in the statement "representations determined by MAMUS behave in a cognitively more plausible manner" also seems a stretch to me.

One issue that the authors should discuss is whether such representations hold any extra value over contextual representations like multilingual Elmo, BERT etc. For instance, why would someone use MAMUS representations instead?

**Experience Assessment:**

I have published in this field for several years.

**Review Assessment: Checking Correctness Of Derivations And Theory:**

I assessed the sensibility of the derivations and theory.

**Review Assessment: Checking Correctness Of Experiments:**

I assessed the sensibility of the experiments.

**Review Assessment: Thoroughness In Paper Reading:**

I read the paper at least twice and used my best judgement in assessing the paper.

---

> ### Author Response · Authors · 2019-11-12
> **Answers to Review #3**
>
> We would like to thank the reviewer for the general interest in our work and the constructive feedback provided.
>
> We regard the stability regarding the number of the nonzero coefficients per word forms characterizing MAMUS a useful property, because it means that one can reliably anticipate the level of sparsity that would be induced by a certain choice of the regularization hyperparameter \lambda. Inspecting the average number of nonzero coefficients per word forms in the case of MultiSparse (or BiSparse), we found much higher fluctuations in the sparsity levels obtained for the same choice of hyperparameters.
>
> Inspired by the question related to the potential usage of multilingual contextual representations, we conducted further XNLI experiments when relying on multilingual BERT representations as inputs. The relative performance of the models trained over MAMUS representations is above 90% to those that are built on top of multilingual BERT embeddings. This suggests that MAMUS representations can serve as a viable alternative to the application of the computationally more demanding contextualized representations.
> For further details, please  refer to the revised version of the paper.

---

### Official Review · AnonReviewer4 · 2019-10-31
**Official Blind Review #4**

**Rating:** 6

**Review:**

This paper describes a method to build sparse multilingual word vectors that is designed to scale easily to many languages. The key idea is to pick one language to be the source language, and then to build word embeddings and then a sparse dictionary + sparse coefficients  for that source language monolingually. Other target languages then first align their embedding to the source using a seed list of translations and standard techniques, and then determine their sparse coefficients based on the fixed source sparse dictionary. This latter process is a convex optimization, which improves efficiency and stability. The method is tested with an established correlation-based intrinsic metric (QVEC-CCA) as well as by using the multilingual embeddings to project systems for cross-lingual document classification, dependency parsing and natural language inference.

The core idea of this paper is substantially simpler than the method it compares to (BiSparse), which jointly optimizes dictionaries, coefficients and cross-lingual coefficient alignment. So, the question that immediately comes to mind for me is, how much accuracy am I giving up for improved scalability to multiple languages? This could be easily tested for the two language case, where BiSparse could be directly compared without modification to their proposed method. Instead, BiSparse is modified to fit scale to the multilingual setting (becoming MultiSparse), but since it shares the constraint that the source dictionary and coefficients are fixed, it has already lost a lot of the power of joint optimization. I think the paper would be stronger with a two-language experiment where we would expect the proposed method to lose to BiSparse, but we could begin to understand what has been given up for scalability.

I also wonder how relevant bilingual word embeddings are in a world of multilingual BERT and similar approaches. It would be interesting to know how cross-lingual embedding-in-context methods would do on the extrinsic evaluations in this paper, though I also acknowledge that this could be considered beyond the scope of the paper.

Otherwise, this is a fine paper. It is well written and easy to follow. The experiments look sane, and the inclusion of both intrinsic and extrinsic tasks makes them fairly convincing. I have only a few remaining nitpicks:

(1) As someone relatively unfamiliar with multilingual (as opposed to bilingual) word embedding research, it wasn’t clear to me how the experiments described here tested the multilinguality (as opposed to bilinguality) of the embeddings. It would be nice to provide an explanation for why (beyond the obvious efficiency gains) one couldn’t just do the necessary language pairs with bilingual methods for these experiments. And if one could perform the tests with bilingual methods, they should be included as baselines.

(2) The discussion of MultiSpare hyper-parameter tuning appears in the Monolingual Experiments section, leading me to wonder what target languages were used for this tuning process.

(3) In the second-last paragraph of 3.2.2, there is a sentence fragment that ends in “nonetheless their multiCluster and multiCCA embedding spaces contain no embeddings for”

(4) The last paragraph before the Conclusion also feels like a fragment, or like two sentences have been spliced together: “We have detailed the differences to Upadhyay et al. (2018) extends the previous work by incorporating dependency relations into sparse coding.”

There are also several places where periods seem to have been left out (such as immediately before the above sentence).

**Experience Assessment:**

I have read many papers in this area.

**Review Assessment: Checking Correctness Of Derivations And Theory:**

N/A

**Review Assessment: Checking Correctness Of Experiments:**

I carefully checked the experiments.

**Review Assessment: Thoroughness In Paper Reading:**

I read the paper at least twice and used my best judgement in assessing the paper.

---

> ### Author Response · Authors · 2019-11-12
> **Answers to Review #4**
>
> We would like to thank the reviewer for the insightful review and the useful suggestions for improving the quality of the paper.
>
> We are planning to report evaluation on QVEC-CCA when using BiSparse instead of MultiSparse. There are two difficulties we have encountered regarding this experiment.
> As BiSparse involves the optimization of nearly twice as much parameters as MultiSparse (since it learns representations for the source and target languages at the same time), training BiSparse representations for a single pair of languages takes more than 50 hours.
> Furthermore, as it turned out, using the same hyperparameters in the BiSparse setting could result in substantially different results.
> For instance, in the case of Italian embeddings, the BiSparse representations contained more than 25 times as many nonzero coefficients as opposed to the representations obtained by MultiSparse, 1100+ and 44.3, respectively.
> This sensitivity of BiSparse for the choice of the regularization hyperparameters, together with its slow running time makes the conduction of the proposed experiment rather cumbersome, given that we would like to compare BiSparse representations of similar sparsity level to the previously calculated MutliSparse representations.
>
> We found the comment on contextual word embeddings a very inspiring one. Hence we conducted further XNLI experiments when relying on multilingual BERT embeddings. We revised the paper with the additional comparisons of multilingual BERT and Mamus. In short, the relative performance of the models that use Mamus instead of multilingual BERT is above 90%. For more details, please refer to the revised version of the paper.
>
> Finally, the revised paper also addresses the further points raised in the review.

---

> > ### Comment · AnonReviewer4 · 2019-11-15
> > **Thanks!**
> >
> > Thanks very much for adding these two experiments: for adding multilingual BERT to the paper and discussing the relation to BiSparse in the comments above (and presumably later adding it to the paper). I think the conclusion that you actually outperform BiSparse really helps strengthen the paper, and I greatly  appreciate your honesty and straightforwardness with the multilingual BERT result.
> >
> > You have definitely addressed all of my major questions.

---

> ### Author Response · Authors · 2019-11-14
> **Comment related to comparison to BiSparse**
>
> As we noted in our general answer, we have experienced that using the same hyperparameters for regularization in the case of BiSparse as for MultiSparse resulted in subpar results in terms of the level of sparsity.
>
> After increasing the monolingual hyperparameters from $\lambda_s=\lambda_t=2$ to $\lambda_s=\lambda_t=5$, we managed to obtain representations relying on BiSparse that behave comparably to MultiSparse as well.
> We created BiSparse representations for the language pairs English--Italian and English--Danish, obtaining approximately 3% of the coefficients becoming nonzero by the end of the optimization.
>
> Evaluating the BiSparse representation by QVEC-CCA, we obtained the scores of 0.602, 0.789/0.786, 0.585 for Danish, English and Italian, respectively.
> We now have two scores for English this time (0.789/0.786) since BiSparse created separate representations for English when jointly trained with Danish and Italian as well.
> MultiSparse on the other hand obtained QVEC-CCA scores as follows: 0.612, 0.808 and 0.596 for the same languages.
>
> This means that MultiSparse performs comparably (or even better) to BiSparse based on their QVEC-CCA scores. We think that this could potentially be explained by the fact that BiSparse tries to optimize a more difficult objective -- as it involves the joint optimization of two non-convex problems -- hence finding a good solution is more difficult in that case.

---

### Decision · Program_Chairs · 2019-12-19

**Decision:**

Accept (Poster)

**Comment:**

This paper describes a new method for creating word embeddings that can operate on corpora from more than one language.  The algorithm is simple, but rivals more complex approaches.

The reviewers were happy with this paper.  They were also impressed that the authors ran the requested multi-lingual BERT experiments, even though they did not show positive results. One reviewer did think that non-contextual word embeddings were of less interest to the NLP community, but thought your arguments for the computational efficiency were convincing.

---

> ### Public Comment · ~Gladis_Ne_Limes1 · 2023-08-15
> **re**
>
> In the intricate world of higher education applications, the value of adept Statement of Purpose (SOP) writing services is like discovering a compass for success. Beyond mere words, these services craft narratives that breathe life into aspirations and experiences. As an aspiring student navigating the competitive landscape https://www.sopwriting.org/, I've realized that SOP writing services are more than just a convenience; they're strategic allies. They understand the art of presenting one's journey with authenticity and impact, helping applicants stand out in the admissions process. With dreams to pursue and opportunities to seize, platforms like these, such as SOPWriting.org, are the guiding light for articulating our academic path. Here's to SOP writing services that empower us to share our story with passion and purpose!